

# Long-term dynamics of monoterpene synthase activities, monoterpene storage pools and emissions in boreal Scots pine

Anni Vanhatalo[1], Andrea Ghirardo[2], Eija Juurola[1,3], Jörg-Peter Schnitzler[2], Ina Zimmer[2], Heidi Hellén[4], Hannele Hakola[4], Jaana Bäck[1]

5 [1]Institute for Atmospheric and Earth System Research / Forest Sciences, Faculty of Agriculture and Forestry, P.O. Box 27, FI-00014 University of Helsinki, Finland
[2]Helmholtz Zentrum München, Research Unit Environmental Simulation, Institute of Biochemical Plant Pathology, D-85764Neuherberg, Germany
[3]ICOS ERIC Head Office, Erik Palménin aukio 1, FI-00560 Helsinki, Finland
10 [4]Finnish Meteorological Institute, P.O. Box 503, FI-00101 Helsinki, Finland

*Correspondence to*: Anni Vanhatalo (anni.vanhatalo@helsinki.fi)

**Abstract.** Seasonal variations in monoterpene emissions from Scots pine (*Pinus sylvestris*) are well documented, and emissions are often shown to follow the incident temperatures due to effects on compound volatility. Recent studies have 15 indicated a link between monoterpene emissions and physiological drivers such as photosynthetic capacity during needle development. The complex interplay between the dynamic changes in the biosynthetic capacity to produce monoterpenes and the temperature-dependent evaporation process of volatiles from internal storage reservoirs has not yet been studied under field conditions.

Here, we analysed the relationships between needle monoterpene synthase activities, endogenous monoterpene storage pools 20 and monoterpene emissions of needles in two consecutive years at a boreal forest site in Finland.

The results showed changes in the monoterpene synthase activity of needles, linked to seasonality and needle ontogenesis, while the pool of stored monoterpenes (about 0.5% of dry weight) did not change considerably as a function of needle aging. Monoterpene emissions did not correlate directly with enzyme activity or the storage pool size. We observed notably high plant-to-plant variation in the biosynthesis rates of individual monoterpenes, which did not reflect the storage compound 25 mixture. The enzyme activity producing δ-3-carene was only present in the first months after needle flushing, and decreased with needle age, whereas δ-3-carene was abundant in the endogenous monoterpene pool and dominated the needle emissions. This study emphasizes the seasonal, developmental and intraspecific variability of monoterpene biosynthesis and storage, and calls for more in-depth analyses to reveal how such complex interaction affects monoterpene emissions from pine needles in boreal forests.

30





## 1 Introduction

The evergreen foliage of conifers needs to acclimate to severe stresses under boreal winter conditions, including low minimum temperatures, low light availability and repeated freeze–thaw cycles. This acclimation is manifested in both structural and metabolic adjustments of needles. Seasonal dynamics in many plant processes creates strong variations in metabolic pools that

enable the needles to remain viable and retain their functional capacity when conditions improve (e.g. Porcar-Castell et al., 2008; Ensminger et al., 2004, 2006). The spring dehardening of coniferous trees is closely linked to physiological changes related to the onset of growth, whereas hardening in the autumn results from the gradual downregulation of cellular metabolism, largely triggered by changes in temperature and the light environment (Hänninen and Tanino, 2011). In addition to primary metabolism related to growth and development, the secondary metabolism of needles, including the synthesis of

volatile compounds, also shows a seasonal pattern that reflects their physiological state (e.g. Fischbach et al., 2002; Jaakola and Hohtola, 2010).

In coniferous plant species, volatile terpenes are produced in all tissues (needles, sapwood, bark, roots) and stored either in specialized terpene storage structures, the resin ducts (Loreto and Schnitzler, 2010), or in non-specific storage pools, for example cell membranes (Niinemets and Reichstein, 2002; Niinemets and Reichstein 2003ab; Ormeño et al., 2011). The

regulation of terpene biosynthesis in needles is a complex process controlled by the availability of carbon substrates, as well as by the energy status of the cell (energy and redox equivalents) and key regulatory enzyme activities (Bohlmann et al., 1998; Fischbach et al., 2002; Dudareva et al., 2004; Ghirardo et al., 2014; Wright et al., 2014). On the other hand, the turnover rates of storage pools depend on environmental constraints (primarily temperature) and on physiological or stress-related processes (e.g. filling of the resin duct storage; herbivory-induced plant defence responses). However, understanding of synthesis and

storage pool dynamics is rather limited. Labelling experiments under laboratory conditions have revealed that in many conifer species, monoterpene biosynthesis in needles is closely linked to the incident photosynthetic carbon supply. It has been shown that 30–60% of emitted monoterpenes originate from recently fixed carbon, comprising the so-called *de novo* emissions, as opposed to the emissions from permanently stored pools (Loreto et al., 2000b; Ghirardo et al., 2010; Ghirardo et al., 2011; Ghirardo et al., 2014).

Large seasonal variations in volatile organic compound (VOC) emission rates and also in the blend of emitted compounds have often been reported from coniferous trees such as Scots pine (*Pinus sylvestris* L.) (e.g. Janson, 1993; Komenda and Koppmann, 2002; Tarvainen et al., 2005). In the summer, VOC emissions are generally highest due to correspondingly high temperatures, but under boreal conditions, the monoterpene emission rates of Scots pine foliage are also quite high in spring, whereas sesquiterpenes are mainly emitted during the summer period (e.g. Tarvainen et al., 2005; Hakola et al., 2006; Aalto

et al., 2014). One hypothesis for such a high seasonal variability is that volatile terpenes may, in addition to their multiple other functions, protect cells from excess energy during periods when photosynthesis is depressed (e.g. Loreto and Velikova, 2001; Owen and Penũelas, 2005; Loreto and Schnitzler, 2010; Velikova et al., 2014; Aalto et al., 2015). In addition to the seasonal fluctuations, high levels of variation exist between and within trees. Buds and young growing needles are a significant source



of VOC emissions during spring dehardening and shoot elongation, and their emission rates can be 1–2 orders of magnitude higher than from older needle age classes (Aalto et al., 2014). Moreover, individual pine and spruce trees produce and emit conspicuous, tree-specific monoterpene blends (Schönwitz et al. 1990a; Bäck et al., 2012; Hakola et al., 2017).

As previously shown in the case of high emissions from emerging foliage (Aalto et al., 2014), phenology is an important driver for seasonal monoterpene dynamics (e.g. Wiß et al., 2017), not only in deciduous plant species but also in evergreens, which retain their foliage for several years. In evergreens, the development of new buds and foliage occurring in spring is characterized by conspicuously high emissions of monoterpenes, methanol and some other VOCs (Aalto et al., 2014). However, long-term studies clarifying the seasonality of monoterpene production and storage in evergreen foliage are scarce, and their correlations with needle emissions have not yet been studied under field conditions. Fischbach et al. (2002) detected strong changes in holm oak (*Quercus ilex*, non-storing species) monoterpene synthase activities as leaves developed and aged. Thoss et al. (2007) observed a chemotype-specific change in the relative composition of monoterpene storage in developing Scots pine needles. However, no consistent seasonal response of stored terpene concentrations was found in Aleppo pine (*Pinus halepensis* Mill.) or three other Mediterranean woody plants in response to drought and warming (Llusià et al., 2006). Litvak et al. (2002) found no relationship between monoterpene storage pool sizes and synthase activities of Douglas fir (*Pseudotsuga menziesii*). Therefore, a reasonable question is how the phenology, seasonality and environmental stimuli affect the production, storage and emission of monoterpenes, and to what extent they are interrelated.

The present work was designed to comprehensively examine the linkages and dynamics of seasonal monoterpene emissions with the corresponding *in vitro* enzyme activities and sizes of storage pools in Scots pine needles *in situ* in a boreal forest. Because needle development (flushing, maturation and gradual ageing) was anticipated to affect the production, storage pools and emissions of monoterpenes, we followed the same branches over two consecutive years to examine possible relationships between the developmental state of needles, monoterpene emission rates, storage pool sizes and monoterpene synthase activities.

## 2 Materials and methods

### 2.1 Site description

The samples were collected from the SMEAR II measurement station (Station for Measuring Ecosystem-Atmosphere Relations) in southern Finland, in a 50-year-old Scots pine forest during 18 months between winter 2009 and summer 2010. The site is located in a managed boreal forest (61°51'N, 24°17'E, 181 m a.s.l.). The forest was regenerated by seeding after a prescribed burning in 1962. The site has been thinned once, in 2002, its density is 1,075 trees ha$^{-1}$ and its leaf area index is 3.5 m$^2$ m$^{-2}$ (Ilvesniemi et al., 2009). The annual mean temperature at the site is 3.5 °C and the annual precipitation is 711 mm (Pirinen et al., 2012).

The height of dominant trees is 18 m, with a breast height diameter of 20 cm. The typical annual tree stem growth rate is 8 m$^3$ ha$^{-1}$ (Vesala et al., 2005). The station is equipped with a comprehensive setup for continuous monitoring of tree and forest floor



gas exchange and the relevant environmental, atmospheric chemistry and physical variables (for a detailed description, see Hari and Kulmala, 2005).

## 2.2 Sampling

The temporal patterns of monoterpene emissions, synthesis and pools were analysed with a repeated sampling design from

four Scots pine trees (#1 to #4) located a few metres from each other. Needle samples were collected from sun-exposed healthy upper canopy branches of the trees, which were accessible from a scaffolding tower. On each sampling occasion, six healthy needle pairs from three branches (whorls 3 to 5, about 1–2 m below treetop) were collected and pooled as one sample per tree. Samples from two needle age classes, formed during summer 2008 (hereafter referred to as *2008 needles*) and 2009 (*2009 needles*), were collected separately and wrapped in aluminium foil. The samples were immediately frozen in a container of

liquid nitrogen and kept at -80 °C until grinding under liquid nitrogen cooling for further biochemical analysis. Two trees (#3 and #4) were intensively sampled between February 2009 and July 2010, generally 1–3 times per month, with the shortest sampling interval occurring during the transient developmental phase in spring (42 sampling days in total, Fig. 1 and Table 1). Additionally, two other trees (#1 and #2) were sampled four times between September 2009 and June 2010 to better understand the tree-to-tree variability in the measured parameters.

Emissions from all four trees were tested before the campaign in order to examine whether they differed from each other in their emission spectrum (see Bäck et al., 2012). Monoterpene emissions were measured in August 2009 from all four trees. Due to limited resources, this was the only time period when all trees could be sampled at the same time. Tree #3 was monitored more intensively: its emissions were measured a total of 15 times between February 2009 and June 2010. The emission monitoring was always carried out from the same upper canopy branch with a transparent FEP (fluorinated ethylene propylene)

foil-covered dynamic flow-through shoot chamber (volume approximately 6 l, see Fig. 2A; further details e.g. in Hakola et al., 2006). The whole shoot, i.e., both 2008 and 2009 needles as well as woody twigs, was enclosed in the chamber. The dry weight of the enclosed needles was about 3 g (final DW was determined later by drying at 80 °C after all measurements had been completed). The terminal bud had been removed well before the first sampling to avoid emissions from growing tissues and changes in the biomass in the enclosure between measurements. Emitted monoterpenes were collected for 60 min in adsorbent

tubes filled with Tenax-TA/Carbopack-B, with flow rates of approximately 4 l min$^{-1}$ through the chamber and 60 ml min$^{-1}$ through the adsorbent tubes. The air entering the chamber passed through a $MnO_2$ ozone scrubber and an active charcoal cartridge. Photosynthetically active radiation (PAR) and the temperature of the shoot chamber were measured with quantum sensors (LI-190SZ, LI-COR Biosciences, Lincoln, USA) and thermocouples, respectively.

Needle and shoot length growth was measured from adjacent trees at the same site, and the data were utilized to determine the

needle development. For the purpose of analysis, needle age (days) was set to zero on the day when needles attained 50% of their final length. This was 27 June 2008 and 29 June 2009 for the 2008 and 2009 needles, respectively. Figures 2B–E present the growth of the shoot and needles in May (when the growth of shoots had just started, but new needles had not yet emerged),





June (the growth of new needles had initiated), July (needles already existed and were used for sampling) and August (fully mature shoot and needles).

## 2.3 Analyses of monoterpene synthase activities and storage pool levels

Monoterpenes stored within the needles (endogenous monoterpene contents) and *in vitro* enzyme activities of monoterpene synthases (MTS) were analysed from the sampled needles of four pines. The analysis of monoterpene storage pools followed the methods introduced in Fischbach et al. (2000, 2002) and further developed in Ghirardo et al. (2010) with the following changes: after pentane extraction of ground frozen samples, 1 µl of the sample was directly injected and analysed using a thermo-desorption (TD) unit (TDU, Gerstel GmbH) coupled to a gas chromatograph–mass spectrometer (GC-MS) (GC type: 7890A, MS type: 5975C inert XL MSD with triple axis detector, both from Agilent Technologies, Palo Alto, CA, USA) using a 5% Phenyl 95% dimethyl arylene siloxane capillary column (60 m x 250 µm x 0.25 µm DB-5MS + 10 m DG, Agilent Technologies). The TD-GC-MS was run as described previously (Ghirardo et al., 2012).

Analysis of *in vitro* MTS activities was carried out in the same way as described in Ghirardo et al. (2012). Briefly, proteins were extracted and successively incubated for 60 min with non-polar, polydimethylsiloxane (PDMS)-coated stir bars (twisters, film thickness of 0.5 mm, Gerstel GmbH) together with the enzyme substrate geranyl diphosphate (GPP). Enzymatically produced monoterpenes were trapped from the aqueous reaction solution by the twisters, and the removal of the stir bars terminated the assays. After rinsing in de-ionized water, the twisters were analysed using TD-GC-MS. Each sample was analysed with three technical replicates. Enzyme activities were only assessed for the samples having protein contents higher than 0.1 mg ml$^{-1}$, and were otherwise recorded as missing values. Calibration was achieved by injecting pure substances in hexane at different concentrations (6–600 pmol µl$^{-1}$) as previously described (Kreuzwieser et al., 2014).

A total of 12 different monoterpenes and their derivatives could be detected from the extracts: α-pinene, camphene, sabinene, β-pinene, β-myrcene, δ-3-carene, geraniol, linalool, linalool oxide, trans-β-ocimene, γ-terpinene and tricyclene. Most of these compounds, however, were only present as traces and therefore only the major monoterpenes, α-pinene, β-pinene, camphene and δ-3-carene, together accounting for >90% of the storage pool, were included in the subsequent analysis. Both assays, (i) analysis of storage pools and (ii) MTS activities, were performed with frozen needle material (fresh weight, FW). For comparison with monoterpene emission rates, the data were normalized to dry weight (DW) using the DW/FW ratios of 0.37 and 0.48 for the newly formed needles and all other needles, respectively (ratios obtained from corresponding measurements of adjacent trees).

## 2.4 Emission analysis

The adsorbent tubes were analysed in the laboratory using a thermal desorption instrument (Perkin-Elmer TurboMatrix 650, Waltham, USA) attached to a GC (Perkin-Elmer Clarus 600, Waltham, USA) with a DB-5MS (60 m, 0.25 mm, 1 µm) column and a mass selective detector (Perkin-Elmer Clarus 600T, Waltham, USA). Five-point calibration was performed using liquid





standards in methanol solutions. The detected monoterpenes were α-pinene, camphene, β-pinene, δ-3-carene, p-cymene, limonene and terpinolene. The method has been described previously, e.g. in Hellén et al. (2012).

The emission rates (E, mg kg⁻¹ DW h⁻¹) were calculated using the concentration difference between the air entering and leaving the chamber as follows:

$$E = [(C_{out} - C_{in})F]/M_{DW}, \qquad \text{(Eq. 1)}$$

where $C_{out}$ and $C_{in}$ are the monoterpene concentrations (g l⁻¹) of outgoing and incoming air, respectively, F (l h⁻¹) is the flow through the chamber and $M_{DW}$ is the dry needle mass (g) enclosed in the chamber. The monoterpene emission potential

(standard emission factor) was obtained using the equation by Guenther (1997) with a β value of 0.09 K⁻¹ and a standard temperature of 30 °C. Results are given in mg kg⁻¹ DW or mg kg⁻¹ DW h⁻¹ to enable comparisons between synthase activity, storage and emissions.

**2.5 Ancillary data from the site**

To explain the variance in monoterpene dynamics, a large amount of auxiliary data from the same measurement site was

employed. Pine foliage net carbon assimilation and transpiration is continuously monitored at the site with automated shoot enclosures, as described e.g. in Altimir et al. (2002) and Aalto et al. (2015). For this study, running averages were calculated of daytime (PAR > 50 μmol m⁻² s⁻¹) carbon assimilation and transpiration for one week preceding needle sampling. Needle and shoot length growth were measured from photographs 1–3 times per week over the growing period and interpolated for the remaining days as presented in Aalto *et al.* (2014). Needle age was calculated as the time (days) since half of the final

needle length had been acquired. A proxy for needle photosynthetic acclimation (S) was calculated with the following dynamic model (Mäkelä et al., 2004):

$$\frac{dS}{dt} = \frac{1}{\tau}(T - S), \qquad \text{(Eq. 2)}$$

where T is the daily mean air temperature and τ is a time constant (h). The parameter S was calculated with two time constants,

60 h and 200 h for S and S2, respectively.

Air temperature at a height of 4.2 m was measured with a ventilated and shielded Pt-100 sensor and soil temperature with thermistors (Philips KTY 81-110, Philips Semiconductors, Eindhoven, the Netherlands) at five spots. The daily air temperature range was calculated as the difference between daily minimum and maximum temperatures. The temperature sum was calculated as the annual cumulative temperature sum of daily mean air temperatures exceeding 5 °C. The soil volumetric water

content (m³ m⁻³) in the mineral soil B horizon was recorded with time domain reflectometers (TDR 100, Campbell Scientific Inc., Logan, USA) and spatially averaged over five plots at the site. Photosynthetically active radiation (PAR, μmol m⁻² s⁻¹) was monitored above the canopy with an LI-190SZ quantum sensor (LI-COR Biosciences, Lincoln, USA). Snow depth was




recorded once a week as an average of five plots at the site. Precipitation was monitored with a tipping bucket rain gauge (Vector ARG-100, Vector Instruments, Rhyl, UK). A rain indicator showed whether there had been rain during the sampling day. The weather conditions (daily mean, minimum and maximum air temperature, daily precipitation and snow depth) during the sampling period were measured as part of the standard measurement setup at the station and are illustrated in Figure 1.

## 2.6 Statistics

Principal component analysis (PCA) was employed to assess whether the variations in MTS activity, storage and emission could be attributed to changes in physical and physiological conditions, such as seasonal changes in the weather (spring–summer–autumn), needle ontogenesis (aging of the needles) and physiological parameters (needle and shoot growth, net $CO_2$ assimilation and transpiration rates). The tree-to-tree variation was not included in the PCA, as the datasets for different trees were not equal in extent. Thus, the analysis was only applied to the dataset of tree #3 plus ancillary data. Datasets from 2009 and 2010 were analysed separately in order to better understand the dynamics related to needle age. The correlation between monoterpene emissions, storage pools, MTS activities (compound-specific), climatic variables, gas exchange and tree physiology data (Table 2) was evaluated using the PCA tools of SIMCA-P (v13, Umetrics, Umeå, Sweden). Thereby, established procedures to analyse MS data were followed as reported previously (Ghirardo et al., 2005; Ghirardo et al., 2012; Velikova et al., 2015). Overall, the matrix was formed by 31 x 36 (variables x observations). Prior to PCA analysis, all variables were logarithmically (log10) transformed and centred, and each type of data was block-wise scaled with 1 sd$^{-1}$. Calculated significant principal components were validated using 'full cross-validation', with the 99% confidence level of parameters and 7 as the number of cross-validation groups.

## 3 Results

### 3.1 Needle age and seasonal effects on monoterpene synthase activities, storage and emissions

We examined in detail the effect of seasonality and needle age on monoterpene synthase (MTS) activity, monoterpene storage pools and emissions, and found that the MTS activities were highly dependent on needle age and season (Fig. 3, 4). Specifically, the youngest needles, i.e. the needles born in 2009 and measured in their first summer and autumn, showed high *in vitro* MTS activities (Fig. 3A, 4A). In particular, δ-3-carene dynamics appeared to be strongly related to needle age: this compound was synthesized in greater amounts in the needles only during their growth and maturation (Fig. 4). The highest MTS activities were measured in the early spring (February–March), independent of needle age (Fig. 3A). A gradual decrease in MTS activities was observed in the second summer of the needles (with a similar decrease in needles born in 2008 and 2009 in their second summer). Consistently, the lowest MTS activities (less than 10% of the maximum) were measured in their third year (when needle age was >600 days, i.e. in 2010; Fig. 3A). Exceptions to this age-specific and seasonal pattern were two individual samples with high MTS activities (e.g. 2008 needles sampled on 19 Oct. 2009 and 4 May 2010), possibly indicating



that the MTS activities were occasionally triggered by abiotic or biotic stresses (although the sampled needles were visually intact).

The pool size of the endogenous (stored) monoterpenes remained much more constant throughout the measurement period and was independent of the needle age (Fig. 3B). The mean monoterpene storage pool sizes were $4.50 \pm 1.03$ and $4.86 \pm 1.22$ g kg$^{-1}$ DW for the young (<1-yr-old) needles and the older ones, respectively. Overall, the amounts of stored monoterpenes in needles ranged between 1.9–8.1 g kg$^{-1}$, with only slightly lower values in winter. The storage pools included on average 31% of δ-3-carene (25–42% in trees #3 and #4), but across all measurement dates, α-pinene was the main monoterpene in pools (54–69% in trees #3 and #4). The temporal variation in storage size mostly resulted from variations in α-pinene concentrations. The highest monoterpene emission rates (up to 6 mg kg$^{-1}$ DW h$^{-1}$, Fig. 3C) from intensively sampled tree #3 were observed at the end of June 2009. The lowest emission rates were detected in spring 2010, when the oldest needles in the shoot chamber were already three years old. The highest emission potentials (standardized to 30 °C), on the other hand, were observed in early spring and autumn (Fig. 3D).

Overall, neither the synthesis rates nor the emission rates directly correlated with the size of the monoterpene pools across the whole measured period (Fig. 4). All four major monoterpenes showed high variability in their associations between storage pools and synthase activities or emissions. For a given storage pool size, the MTS activities varied by several orders of magnitude, with highest MTS activities present in <1-yr-old needles (Fig. 4A–D).

Principal component analysis (PCA) was therefore employed to examine the complex link between meteorological and physiological parameters that played a major role in the dynamic changes in MTS activities, storage pools and monoterpene emissions. For both years, the variation in the data could be fairly well described by two main factors: seasonality (principal component 1, PC1) and needle aging (principal component 2, PC2), which together accounted for 60% (2009) and 54% (2010) of the total variation in the dataset (Fig. 5). This was indicated by the clear separation in PC1 of summer (Fig. 5A–B, depicted in red), spring (in white) and autumn (in grey) samples for both 2008 and 2009 needle samples, and by the separation made by PC2 of 2008 needle samples (triangles) from 2009 needle samples (circles) (Fig. 5A–B). Meteorological and physiological data, in particular air temperatures, carbon assimilation, transpiration and needle growth rates, correlated positively with the monoterpene emission rates and negatively with MTS activities (Fig. 5C–D). Within the same year, the MTS activities were significantly and positively correlated with younger needles, and high monoterpene emissions with summer samples (Fig. 5A–D). In 2009, the endogenous monoterpene content was lower in young needles compared to the content in mature needles. Such a difference was absent in 2010, when both needle age classes were already mature, i.e. two and three years old, respectively.

Taken together, the multivariate data analysis revealed a seasonal and ontogenesis-related dependency of emissions, storage and biosynthesis of monoterpenes. Changes in both MTS activity and the monoterpene pool size were related to needle ontogenetic phases. These changes notably occurred during needle development and needle maturation, and were also affected by seasonality.



## 3.2 Tree-to-tree variation

Aiming at analysing the different tree chemotypes, we took needle samples twice from all the four trees that had initially been screened for the monoterpene blend in the emissions, and analysed the MTS activity, storage pools and emissions (only once) from the individual trees from the youngest needle year class (when the needles were 2 and 9 months old). The MTS activity

varied between the trees: it was lowest in needles from tree #2 both in summer and winter, and highest in needles from tree #4 in spring (>4 times higher than in tree #2) (Fig. 6A–B). The storage pool size was relatively stable across all trees and sampling times (Fig. 6C–D). In all four trees, α-pinene was the most abundant monoterpene in the enzyme assay. The proportions of the four main monoterpenes changed little in both MTS assays and in storage pools between summer (Aug.–Sep.) and winter (March) in a given tree (Fig. 6A–D). However, a qualitative tree-specific difference was detected among the stored

monoterpenes: the proportion of δ-3-carene was pronounced (comprising 30–50% of the total monoterpenes) in the storage pools in trees #2 to #4 (Fig. 6C–D), whereas it was almost absent from the enzymatic monoterpene pattern of these trees (Fig. 6A–B). Furthermore, in tree #1, δ-3-carene was completely missing from both MTS and storage.

The trees differed strongly in their monoterpene emission patterns (Fig. 6E). All trees emitted δ-3-carene, but it was most dominant (52%) in tree #2, which also emitted high quantities (31%) of ß-pinene and had the highest emission rate in August.

Tree #1 dominantly emitted α-pinene (74% of overall monoterpene emissions), while this compound contributed only 17%, 24% and 35% to the emission pattern in trees #2, #3 and #4, respectively. Tree #1 was thus classified as a clear 'pinene chemotype', while the other three trees were 'intermediate emitter types' (see Bäck et al., 2012). Interestingly, absolute monoterpene emission rates also varied considerably between the individual trees (Fig. 6E), although the temperatures in 2009 during the sampling period did not vary by more than a few degrees and other conditions did not differ, and no visible damage

was seen in sampled branches. The variation in the standard emission factors for the four trees #1–4 was large, ranging from 3.2 (tree #3) to 119 (#2) mg kg$^{-1}$ DW h$^{-1}$.

## 4 Discussion

Although the seasonal variations in monoterpene emissions from evergreen trees are well documented (e.g., Komenda and Koppmann, 2002; Tarvainen et al., 2005 Hakola et al., 2006), the reasons for this variation are poorly understood. The common

explanation includes the temperature response of emissions, due to the strong role of temperature in physical parameters such as volatility and diffusion (e.g., Guenther, 1997; Niinemets and Reichstein, 2003a; 2003b). This indeed creates seasonal dynamics, which partially but not fully explain the observed emission rates. The present results showed that the potentials to synthetize monoterpene, i.e. enzyme activities, are strongly dependent on needle age and season. However, these monoterpene synthase activities did not correlate with needle emission rates, and neither did the storage pool dynamics, which were virtually

constant throughout the field experiment.

As indicated by the PCA analysis, the interaction between monoterpene synthesis, storage and emissions is complex and dependent on climatic factors (season) and needle age. Therefore, changes in emission rates at the leaf scale are probably a



consequence of much more complex mechanisms than a simple incident temperature proxy (Aalto et al., 2014). This signifies the need to understand the physiological drivers, in addition to the physico-chemical drivers behind emissions.

The possible relationships between monoterpene production, storage and emissions were here investigated using simultaneous measurements of enzyme activities, storage pools and emissions. The results indicate that synthesis, emission rates and storage are mainly decoupled. For instance, high MTS activities did not correspond to large storage pools or high emission rates, while high emissions were not a result of large storage pools. The likely reasons for this are, on the one hand, the large monoterpene storage pools in needles, and on the other hand, the disparity and time lags between production and emissions. Here, we see an analogy with VOC synthesis in *Lamiaceae*: during their early ontogenetic development, synthases actively produce volatiles in leaf glands, but biosynthesis slows down once storage pools in the glands are full (e.g. McConkey et al., 2000).

Our results demonstrate that the monoterpene storage pool can make up ca. 0.2–0.8% of needle dry weight. These concentrations are in a similar range to previous observations from other conifers (Lerdau et al., 1997; Litvak and Monson, 1998; Litvak et al., 2002; Kännaste et al., 2013). Since emissions were three orders of magnitude less than pool sizes, they are unlikely to affect the pool sizes in the short to medium term. Similarly, the effect of MTS activities on pool sizes is also quite minimal in mature needles and in the short term, and thus it did not show up in any correlations. It is known that incident monoterpene emissions simultaneously originate from previously filled and specialized storage pools and from *de novo* synthesis. In Scots pine, *de novo* synthesis can contribute ~60% of the total emissions (Ghirardo et al., 2010), although this percentage must depend on both the rate of *de novo* production, which is light and temperature dependent and under the control of enzyme activities, and the magnitude of the emission rates from storage pools, which are only temperature dependent (Ghirardo et al., 2010; Loreto and Schnitzler, 2010). Changes in *de novo* biosynthesis can be expected to vary under different environmental conditions and, in light of the actual results for enzyme activities, follow seasonal and needle-age dependencies. Consistently, recent findings from the needles of field-grown Scots pines indicate that the ratio of *de novo* to storage pool emissions is not stable, but varies considerably in spring during photosynthetic recovery (Aalto et al., 2015). Our results indicating high MTS activity in young developing needles support this observation of highly dynamic biosynthesis.

The decoupling between monoterpene production and storage pools and emissions was also qualitatively evident: the individual MTS activities did not reflect the monoterpene composition found in the storage pool or in the shoot emission. For instance, δ-3-carene was the second most abundant component in storage pools and also very prominent in emissions, but appeared in much smaller quantities than α-pinene in the MTS activity. Similar decoupling of δ-3-carene has also been found in ponderosa pine (*Pinus ponderosa*) (Harley et al., 2014).

The overall variation in the dataset was best correlated with the season and needle age, as indicated in the PCA analysis. Young needles generally displayed a larger emission capacity than older ones, reflected in their higher MTS activities. Our data indicate that needles retain a high capacity for monoterpene synthesis throughout their first full year of growth, but the MTS activity later sharply declines. The monoterpene concentrations and the relative proportions in Scots pine needles are known to change during the first months of new needles (Thoss et al., 2007), and even though our data do not cover the very first months of needle development, our results support this observation. In particular, the δ-3-carene dynamics appeared to be





strongly related to needle age: this compound was synthesized in greater amounts in the needles only during their growth and maturation. In developing needles of trees #3 and #4, δ-3-carene contributed 11–19% to the total MTS activity, but the percentage declined during the following autumn to a lower level below the detection limit and remained at this level in older needles. This enzyme activity was sufficient to fill and sustain the storage pools: the corresponding proportions of δ-3-carene

in monoterpene storage pools of the new needles in trees #3 and #4 were 38–42%, and later in the autumn 27–34%. Thoss et al. (2007) also reported such a developmental response of δ-3-carene for the first three months after needle emergence. Interestingly, δ-3-carene has been considered as the least toxic monoterpene to bark beetles and their associated fungi (Raffa et al., 2014; Reid et al., 2017). This suggests that the abundance or scarcity of δ-3-carene (i.e. the pine chemotype) could play a key role in the interactions between pines and other organisms.

The stability of the storage pool suggests that the monoterpene storages are filled during the very first weeks or months after leaf emergence (Bernard-Dagan, 1988; Schönwitz et al., 1990). This is logical when considering that one of the main reasons for storing monoterpenes in needles is their protection from herbivory (Langenheim, 1994; Litvak and Monson, 1998; Loreto et al., 2000a), and the youngest needles are particularly susceptible to many insects feeding on the fresh, sugar-rich tissues. This is also in line with the higher MTS activity in the youngest needles.

Our data suggest that the turnover of the permanent storage is a very slow process, and agree with the low rate of incorporation of $^{13}C$ into the storage pool observed after 8 h of $^{13}CO_2$ labelling in Scots pine seedlings (Ghirardo et al., 2010). Earlier, no variations were found in the monoterpene storage composition between different age classes or between seasons in the case of mature Scots pine needles (Thoss et al., 2007). Some monoterpenes may also accumulate in needle surface waxes (Joensuu et al., 2016; Despland et al., 2016). However, needle wax is probably a rather short-term monoterpene storage and may not have

a major impact on the incident emission rates, at least for the main emitted compounds.

The time lag between monoterpene synthesis and emissions can probably range from minutes (*de novo*) to days (permanent pools). The MTS activities could possibly correlate with *de novo* emissions in the short term, but this cannot be addressed with our data, which were collected at approximately bi-weekly intervals. A much finer temporal resolution and labelling experiments should be used to analyse the relationships further. Besides, it is evident that the synthase activities we measured

here reflect the maximum potential of the needle tissue to synthesize monoterpenes under optimal conditions of temperature, pH and saturated substrate availability. Thus, they may not represent *in situ* synthesis processes, and this is a possible reason for the observed decoupling of production and storage/emissions.

As we aimed at a measurement setup with repeated samplings, the challenge was to maintain the extremely delicate monoterpene storage pools as intact as possible and to avoid any induced emission responses (Niinemets et al., 2011).

However, the monoterpene synthase activities varied between and within trees, and some days with exceptionally high MTS activities and emissions were observed. They may be due to certain short-term processes such as transient responses to herbivory or mechanical stress. It is known that handling a pine shoot causes increased monoterpene emissions for a few days (Ruuskanen et al., 2005). In addition, the observed anomalous emission blend of tree #3 in September 2009 could originate from a stress reaction, and such unexplained high emission peaks have also been reported earlier in similar measurement setups



(e.g. Tarvainen et al., 2005). Nevertheless, no visible damage was observed in the needles prior to sampling, and the twigs were not mechanically injured.

Furthermore, we collected the first set of youngest needles in mid-July, at a time when shoot lignification had not been fully completed and needle elongation was still in progress (ca 85% of final length, see Fig. 2). In practice, sampling short shoots

before their lignification is already advanced and causes large wounds and ample resin flow from the wounded twig.

One caveat in the emission measurement method is that the enclosures contained needles from two older year classes, but did not contain the youngest (2009) needles at all. Thus, we cannot separate the effect of needle development and maturation from this data set: almost all emission measurements were performed from already mature needles, and only a fraction of the emissions originated from <1-yr-old needles in the spring and summer of 2009. A part of the emissions may also have

originated from non-needle sources, i.e. the woody tissues of twigs.

This study was the first to address the relationship between monoterpene synthase activities, storage pools and emissions *in situ* in a resin-storing conifer, Scots pine. Our results emphasize the seasonal, developmental and intraspecific variability in monoterpene biosynthesis and storage, and call for more studies to reveal their connections with emissions. The results of this study and forthcoming studies should be applied in the VOC modelling in the future: e.g., the blend of monoterpenes in plant

storage structures should not be used as an estimate of the emission composition.

## 5 Funding

This work was supported by the Academy of Finland Centre of Excellence programme [1118615 and 272041], the Helsinki University Centre for Environment HENVI [470149021], the Nordic Centre of Excellence CRAICC and the COST Action FP0903 'Climate Change and Forest Mitigation and Adaptation in a Polluted Environment'.

## 6 Acknowledgements

We thank Juho Aalto and Janne Korhonen for assisting in sample collection and data analysis. The SMEAR II staff are acknowledged for their maintenance of the measurements and infrastructure.

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





**Tables and figures**

**Table 1. Details of the examined trees and sampling.**

| Tree # | Height (m) | Diameter at 1.3m (cm) | Number of needle samples | Number of emission samples |
|--------|------------|------------------------|---------------------------|-----------------------------|
| 1 | 17.7 | 16.6 | 4 | 1 |
| 2 | 17.9 | 18.4 | 4 | 1 |
| 3 | 17.6 | 17.3 | 21 | 15 |
| 4 | 17.8 | 18.6 | 40 | 1 |

**Table 2. The variables used in PCA.**

| Abbreviation | Variable description |
|--------------|----------------------|
| A | Pine shoot $CO_2$ assimilation |
| E | Pine shoot $H_2O$ transpiration |
| n | Needle growth |
| s | Shoot growth |
| age | Needle age |
| PAR | Daily maximum of photosynthetically active radiation |
| Tmean | Daily mean air temperature |
| Tmax | Daily maximum air temperature |
| Tmin | Daily minimum air temperature |
| Tr | Daily temperature range |
| Ts | Cumulative temperature sum |
| Tsoil | Soil temperature |
| S2 | Proxy for needle photosynthetic acclimation |
| snow | Snow depth |
| rain | Daily precipitation |
| rainy | Rain indicator |
| VWC | Soil volumetric water content |





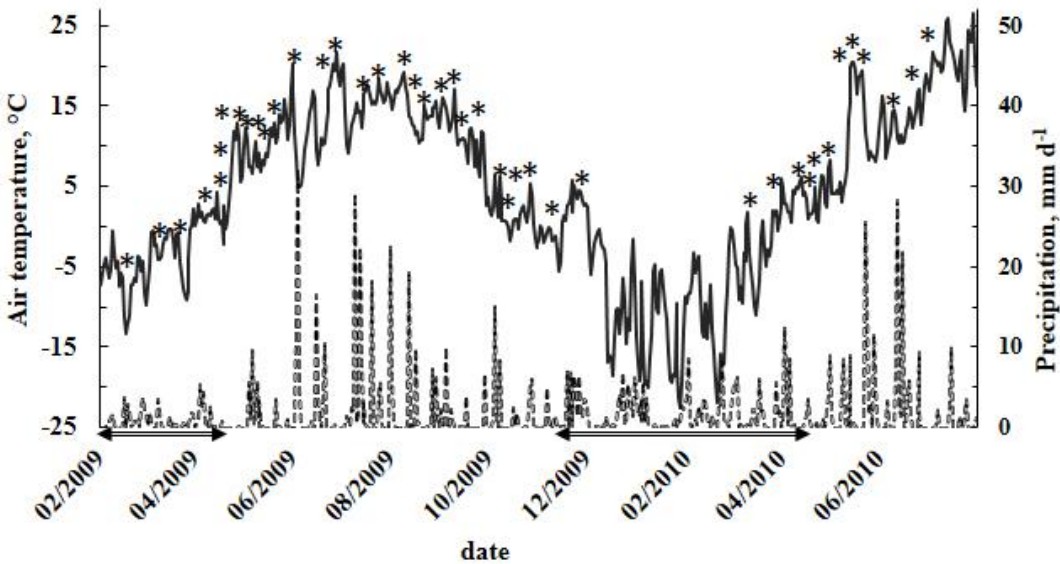

**Figure 1: Daily mean air temperature (solid line) and daily precipitation (dashed line) at the sampling site during the sampling period from February 2009–July 2010. Stars indicate the sampling dates (not all the trees were sampled every time). Horizontal arrows represent the snow cover periods.**

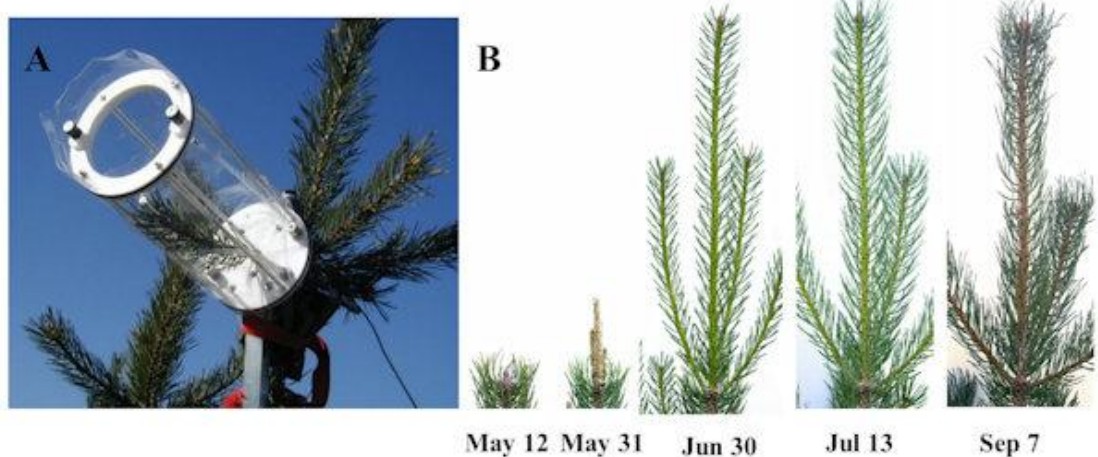

**Figure 2: (a) Emission measurements were performed with a transparent shoot chamber, which is here open between samplings. (b) Development of a new shoot in Scots pine in 2010. Sampling of new needles was started in early July. Photos: Juho Aalto.**



**Figure 3: The synthase activities of α-pinene, β-pinene, camphene and δ-3-carene (sum, a), the storage pools of the same compounds (sum, b), their measured emission rates (sum, c), and emission potentials (standardized to 30 °C) (sum, d) as functions of time. In (a) and (b), the curves show the means ± 1 standard deviation of the samples of different needle age classes (grown in 2008 and 2009)**

5   **collected from trees #3 and #4. Missing error bars represent the dates when data were only available from one tree. In (c) and (d), the emissions originate from a shoot of tree #3.**





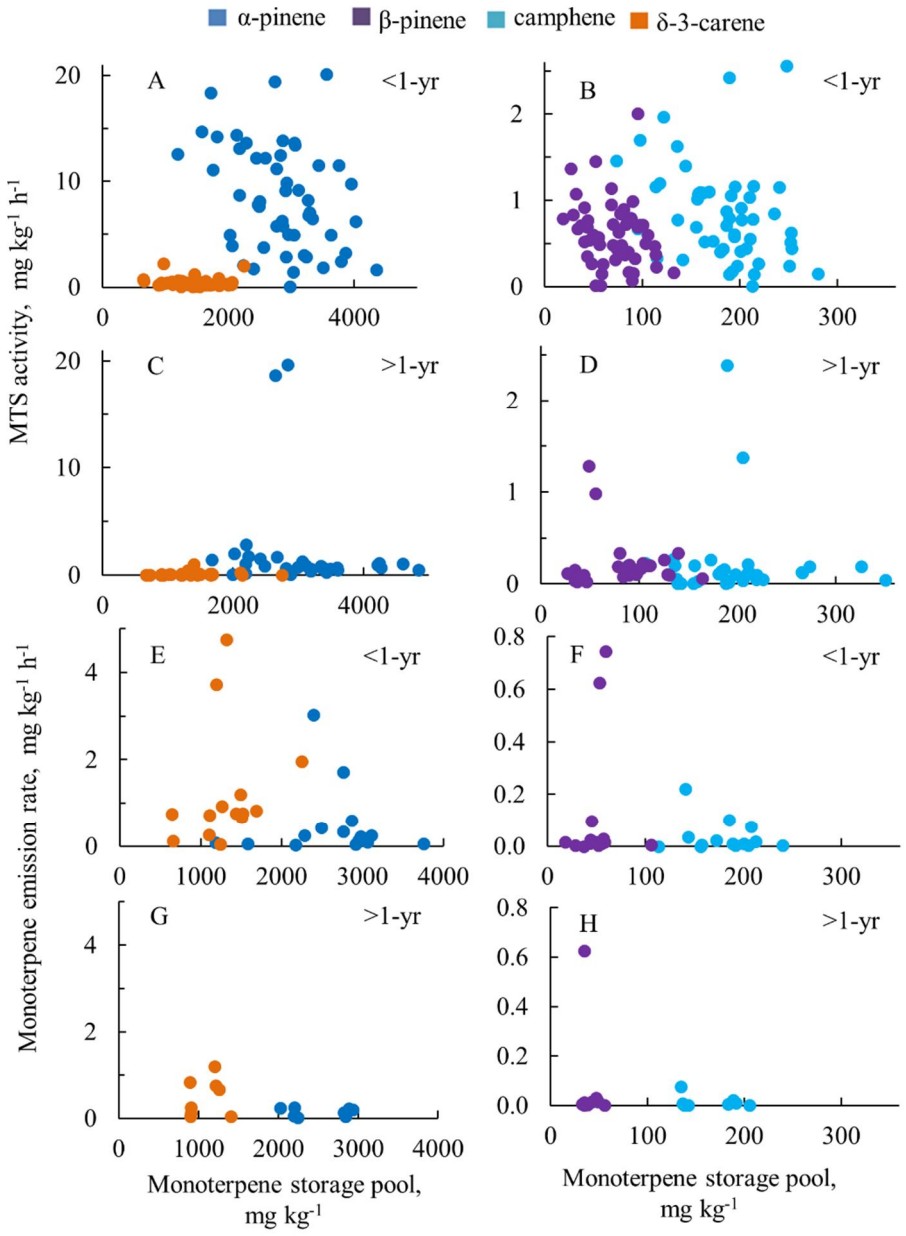

**Figure 4: Relationship between monoterpene storage pools (x-axes) and MTS activities (a–d) and between monoterpene emissions (e–h) in <1-yr-old (a–b, e–f) and >1-yr-old (c–d, g–h) needles. Plots for more abundant compounds (α-pinene and δ-3-carene) on the left and for less abundant compounds (β-pinene and camphene) on the right. Emissions originated from both <1-yr-old and >1-yr-old needles and a twig. Data from trees #3 and #4.**





**Figure 5: Two-dimensional (a–b) score plots and (c–d) scaled and centred loading plots of PCA from monoterpene synthase activities, storage pools, emissions, and meteorological and physiological data collected in 2009 and 2010, respectively. Calculated significant**

5 **principal components were validated using 'full cross-validation', with the 99% confidence levels of parameters (cross-validation groups = 7). The explained variance (as a percentage) and the number of the principal component are given in both x- and y-axes. The ellipses in (a) and (b) indicate the tolerance based on Hotelling's $T^2$ with a significance level of 0.05. From outer to inner, the three circles in (c) and (d) indicate 100%, 75% and 50% explained variance, respectively. Lines in (a) and (c) have additionally been added to indicate the borders between the 2008 and 2009 needle samples and their respective variables.**

10 **Symbol legend (a–b): spring 1 March–15 May (white symbols), summer 16 May–14 Aug. (red) and autumn 15 Aug.–30 Oct. (grey). The 2008 and 2009 needle samples are depicted with triangles and circles, respectively. α refers to α-pinene, β to β-pinene, c to camphene, and δ to δ-3-carene. The other variables are listed in Table 2.**





**Figure 6: (a) and (b) Tree-to-tree variation for 2009 needles in monoterpene synthase activity and (c) and (d) monoterpene storage in late summer/autumn (Aug.–Sep. 2009: a, c, e, approx. two-month-old needles) and in early spring (March 2010: b, d, approx. nine-month-old needles). (e) Monoterpene emission rates were measured from all the studied trees in August 2009.**

