# Peer review of "Long-term dynamics of monoterpene synthase activities, monoterpene storage pools and emissions in boreal Scots pine"

_Biogeosciences, 2018_

## Referee Comment (RC1) · Anonymous Referee #1 · 9 Feb 2018

Comments by Reviewer

General comments: The authors present "Long-term dynamics of monoterpene synthase activities, monoterpene storage pools and emissions in boreal Scots pine". It is important to evaluate the monoterpene emissions in boreal coniferous forest. However, I think that the interpretations of the data to reach conclusions are fine with the exception of very few data points (monoterpene emission data) used in this study. The manuscript must be revised by the authors for possible publication in Biogeosciences.

Specific comments: 1. P4 L23-24: High monoterpene emission rate might be caused by the mechanical disturbance when the terminal bud was removed. Did the authors

check the time required for stabilizing the BVOC emission rate after cutting the terminal bud?

2. P8 L17-29: I think that the number of monoterpene emission data is very small (only a few days). Can the authors justify that this small data set is enough data to make the conclusions presented in this paper?

3. P9 L20-21: "The variation in the standard emission factors was large." The authors did not show the relationship between monoterpene emission rate and temperature. In addition, did authors confirm that whether the coefficients $\beta$ (= 0.09) is valid for monoterpene emission from Scots pine? Please mention.

4. P9 L24: Tarvaine et al., 2005; Hakola et al., 2006 (add ;).

5. Discussion: 1) I think that the amount of the monoterpene precursor (geranyl diphosphate: GPP or GDP) and the activity of monoterpene synthase are involved in monoterpene production and emissions. Please discuss the GPP in leaves.

2) The authors measured monoterpene emissions from Scots pine. $\delta$-3-carene was the dominant compound in Tree #2, whereas $\alpha$-pinene was the dominant compound in Tree #1. The composition of the identified monoterpenes was different for different trees (#1-#4). In addition, large tree-to-tree variations in standard emission factors were observed. However, the tree-to-tree variation was not included in the principal component analysis (PCA). Did authors confirm that whether the total monoterpene emission is valid for PCA analysis? I think that individual monoterpenes (e.g., $\delta$-3-carene and $\alpha$-pinene) should be applied separately in PCA analysis in order to better understand the dynamics related to monoterpene pools and monoterpene synthase activities. Please mention.

---

## Referee Comment (RC2) · Anonymous Referee #2 · 19 Feb 2018

Long-term dynamics of monoterpene synthase activities, monoterpene storage pools and emissions in boreal Scots pine

The end aim of this submission is modelling monoterpene emission based on plant physiology along the growth season and needle development. The relevance of terpene emission in the biosphere-atmosphere interphase is increasingly recognized, however, in fact, the link between terpene biosynthesis, storage and emission is not well understood, either the direct and indirect effects of resource availability, meteorology and carbon fixation on the whole chain of steps leading to monoterpene emission. Authors use a multiple approach combining an intensive field sampling along two years,

evaluation of terpene emission and storage pools and the activity of terpene synthases, with the inclusion of a large number of physiological and environmental variables as potential predictors of terpene emission. This multiple approach is novel and highly valuable, although the sample size for emission is relatively contained. The manuscript is well written, introduction organized in a logical sense and methodology describe the methods with precision in a concise way (but see specific comments). I have just few suggestions with alternative approaches to the data set, and minor comments on the text.

Main comments on results and discussion

I would suggest thinking as in dendroecological studies and try to decouple climate and emission checking the fit of the regression after adding different time lags ... it is likely that meteorology or physiology during the day before could explain better the emission than during the same day. Did you try this? Please, explore it if not. Regarding Fig 5. As a suggestion, if you want mix in the same multivariate model your descriptors (meteo, physiology, etc) and your variables (emission, storage, MTS) I would suggest using a NMDS model or any other than PCA (which is a clearly parametric model for summarizing variables). Alternatively, you could summarize your ancillary properties and descriptors (including in the descriptors this time physiology, MTS and stored terpene pool) with a PCA, and then correlate the axis obtained with the emission values you got at field. A concern (that not flaw) is that MTS activity assay informs of the in vitro maximum potential activity. In vivo terpene production would depend on many factors such as enzyme activity, enzyme concentration, substrate availability, etc. Although you already state this limitation clearly in the discussion, as a suggestion, I miss a more extended discussion about this interesting point. Due to the novelty of your mixed approach, I would ask for incorporate into the last part of your discussion some material about future research, how your methodological approach provided light into future experimental designs and methodologies to be applied to, and what requirements new experiments pursuing your aim must accomplish.

Specific comments P3L14. May be the introduction would gain introducing the role of biotic induced responses as a source of plasticity in the amount and profile of terpenes. Such information would be valuable later in the discussion, as wounds made with the chamber could be a source of emission variability between samplings and plants. P4L15. You report that you tested the four experimental trees before. So, could you explain what was the reason for sampling in those 4 trees with so different emission spectrum instead to increase your sample size focussing your effort in more similar individuals. P4L25. Please report the mass (mg) of Tenax and Carbopack in the traps for allow experiment repetition. P5L5. I would suggest reportting the solvent:sample (d.w.) ratio for a complete description. P5L15. Please, state here the sample size for MTS activity. P6L5. May be I am missing something, but I cannot understand how do you apply this equation because you explain that air entering in the chamber was flowed thru a charcoal trap. P6L10. Please, report what was the range of temperatures for your emission samplings. P21. (fig 5) In order to be consistent with the codes in other figures and in the text, suggest labelling the panels with the age of the needles instead the actual year. I mean "<>1 yr old" instead "2008 needles".

СЗ

---

## Author Comment (AC1) · 27 Mar 2018

**Author response to the Anonymous Referees #1 and #2**

on the manuscript "Long-term dynamics of monoterpene synthase activities, monoterpene storage pools and emissions in boreal Scots pine" in Biogeosciences Discuss., https://doi.org/10.5194/bg-2018-17-RC1, 2018

by Anni Vanhatalo et al.

We thank both Referees for their constructive comments and suggestions to improve the manuscript. You will find below our detailed responses to them in red.

Comments by Reviewer #1

General comments: The authors present "Long-term dynamics of monoterpene synthase activities, monoterpene storage pools and emissions in boreal Scots pine". It is important to evaluate the monoterpene emissions in boreal coniferous forest. However, I think that the interpretations of the data to reach conclusions are fine with the exception of very few data points (monoterpene emission data) used in this study. The manuscript must be revised by the authors for possible publication in Biogeosciences.

Specific comments: 1. P4 L23-24: High monoterpene emission rate might be caused by the mechanical disturbance when the terminal bud was removed. Did the authors check the time required for stabilizing the BVOC emission rate after cutting the terminal bud?

The bud was removed well before the first emission sampling took place, weeks ahead. During all the samplings there was no fresh (soft and thus monoterpene emitting) resin at the bud scar. Earlier studies on debudded Scots pine shoots with VOC monitoring (e.g. Hakola et al. 2006) have shown that the increased emissions cease in shorter time (less than 7 days) than applied here, even in winter conditions. Thus, we believe that the sampled emissions are not induced by the bud removal. To emphasize this, we added a sentence: "As the settling time here was considerably longer than the period of debudding-induced increased emissions in earlier studies (e.g. Hakola et al., 2006), increased emissions were unlikely to occur once the sampling started."

2. P8 L17-29: I think that the number of monoterpene emission data is very small (only a few days). Can the authors justify that this small data set is enough data to make the conclusions presented in this paper?

The PCA analysis was conducted for the tree, which emissions were monitored most intensively (tree #3). For that tree, the number of emission samplings (15, given in the Table 1) was roughly once a month for the whole study period (spring 2009–summer 2010) excluding the winter months (October–February). We believe that this frequency was sufficient to catch the seasonality rather well. Nevertheless, we agree with the reviewer that if possible, more frequent sampling would have been ideal.

3. P9 L20-21: "The variation in the standard emission factors was large." The authors did not show the relationship between monoterpene emission rate and temperature. In addition, did authors confirm that whether the coefficients $\beta$ (= 0.09) is valid for monoterpene emission from Scots pine? Please mention.

The relationships between monoterpene emission rates and air temperatures were not analysed. However, the relationship is well-known from previous literature and thus we did not emphasize it

here. The β coefficient of 0.09 has been applied for Scots pine in numerous earlier studies; e.g. in Hakola et al. (2006), Kivimäenpää et al. (2012), and Aalto et al. (2015), and to allow comparisons with these published data we decided to use the same value. Further, in Tarvainen et al. (2005), the compound-specifically determined β values were close to 0.09. Thus, we think its use is justified. For clarification, we edited the text as follows: "The monoterpene emission potential (standard emission factor) was obtained using the equation by Guenther (1997) with a β value of 0.09 $K^{-1}$ (frequently used value for Scots pine, e.g. in Hakola et al., 2006; Aalto et al., 2015) and --"

4. P9 L24: Tarvaine et al., 2005; Hakola et al., 2006 (add ;).

Sorry for a typo, semicolon added.

5. Discussion: 1) I think that the amount of the monoterpene precursor (geranyl diphosphate: GPP or GDP) and the activity of monoterpene synthase are involved in monoterpene production and emissions. Please discuss the GPP in leaves.

We agree with the reviewer that substrate availability is involved in the productions of (any) enzyme reaction. Isoprenoid biosynthesis occurs in plastids or cytoplasm, and relies directly or indirectly on photosynthesis for the supply of carbon substrates (Lichtenthaler et al., 1997; Phillips et al., 2008). However, the impact of substrate availability on enzyme reaction can vary largely, depending on the pool size of the substrate and the velocity of the enzyme to catalyse its reaction. Conversely to isoprene production, which is under control of both precursors and enzyme activities, monoterpene synthesis is much more dependent on enzyme activities than affected by substrate limitation. This is due to the much lower Michaelis-Menten constant (Km): Km for monoterpene synthase (MTP) is of ~3 µmol L-1 and for isoprene synthase (ISPS) is 500 µmol (e.g. (Zimmer et al., 2000; Tholl et al., 2001). The much lower Km means that the large pool of the substrates GDP, as suggested from other studies (Nogués et al., 2006) is not likely to be the limiting factor of monoterpene emissions. At least this is valid until the photosynthesis is not impaired, as in our field study. Further, since a considerable part of the carbon in monoterpenes originates from other metabolic pathways than directly from photosynthesis, the role of GPP is difficult to assess. Since we did not measure the GDP levels in our samples it is impossible to make a final conclusion on the role of substrate limitation. We would like to refrain from adding discussion on GDP levels in our manuscript since we feel this would be merely speculative.

2) The authors measured monoterpene emissions from Scots pine. δ-3-carene was the dominant compound in Tree #2, whereas α-pinene was the dominant compound in Tree #1. The composition of the identified monoterpenes was different for different trees (#1-#4). In addition, large tree-to-tree variations in standard emission factors were observed. However, the tree-to-tree variation was not included in the principal component analysis (PCA).

The PCA analysis was conducted only for the data from tree #3, where all data (emission rates, pool levels, enzyme activities etc.) were available. Unfortunately, the different dimensions of our datasets did not allow studying the tree-to-tree variation using multivariate data analysis, but we agree with the reviewer that this could have been an interesting point to address with chemometrics.

Did authors confirm that whether the total monoterpene emission is valid for PCA analysis? I think that individual monoterpenes (e.g., δ-3- carene and α-pinene) should be applied separately in PCA analysis in order to better understand the dynamics related to monoterpene pools and monoterpene synthase activities. Please mention.

The PCA analysis was done compound-specifically instead of lumping all the monoterpenes together. This was seen as a better way to study the dynamics and to observe the possible differences between the compounds.
* * *
The end aim of this submission is modelling monoterpene emission based on plant physiology along the growth season and needle development. The relevance of terpene emission in the biosphere-atmosphere interphase is increasingly recognized, however, in fact, the link between terpene biosynthesis, storage and emission is not well understood, either the direct and indirect effects of resource availability, meteorology and carbon fixation on the whole chain of steps leading to monoterpene emission. Authors use a multiple approach combining an intensive field sampling along two years, evaluation of terpene emission and storage pools and the activity of terpene synthases, with the inclusion of a large number of physiological and environmental variables as potential predictors of terpene emission. This multiple approach is novel and highly valuable, although the sample size for emission is relatively contained. The manuscript is well written, introduction organized in a logical sense and methodology describe the methods with precision in a concise way (but see specific comments). I have just few suggestions with alternative approaches to the data set, and minor comments on the text.

Main comments on results and discussion

I would suggest thinking as in dendroecological studies and try to decouple climate and emission checking the fit of the regression after adding different time lags ... it is likely that meteorology or physiology during the day before could explain better the emission than during the same day. Did you try this? Please, explore it if not.

Thank you for this interesting suggestion. Unfortunately, our sample collection was done on campaign wise manner with predetermined sampling dates, and the data did not allow such analysis in detail. However, there were some variables in the PCA that represented the meteorological dynamics, in particular the S parameter with time constants of 60 and 200 hours as well as the temperature sum over the growing period. They both include not only the incident conditions, but also the history effect.

Regarding Fig 5. As a suggestion, if you want mix in the same multivariate model your descriptors (meteo, physiology, etc) and your variables (emission, storage, MTS) I would suggest using a NMDS model or any other than PCA (which is a clearly parametric model for summarizing variables). Alternatively, you could summarize your ancillary properties and descriptors (including in the descriptors this time physiology, MTS and stored terpene pool) with a PCA, and then correlate the axis obtained with the emission values you got at field.

Thank you for this suggestion. However, we decided to select the PCA method to assess if the variations in MTS activity, storage and emission could be attributed to changes in environmental stimuli together and not one-by-one. As our results show, the variation in MTS activity, storage and emission could be explained to great extent with two main drivers, seasonality and needle age.

A concern (that not flaw) is that MTS activity assay informs of the in vitro maximum potential activity. In vivo terpene production would depend on many factors such as enzyme activity, enzyme concentration, substrate availability, etc. Although you already state this limitation clearly in the discussion, as a suggestion, I miss a more extended discussion about this interesting point.

The point is interesting indeed. Monoterpene production depends on the substrates, energy availability and enzymatic factors. However, our sampling protocol of field-grown trees was optimized to maintain the sample shoots as intact as possible for the whole study period, with the consequence of limiting the sample size to a few needle pairs. Thus, it allowed us only to analyse the enzyme activities instead of larger assays of substrates and precursors. Any further discussion of the other factors would therefore be fully speculative. This same point is discussed also in our response to the Referee #1, please see above.

Due to the novelty of your mixed approach, I would ask for incorporate into the last part of your discussion some material about future research, how your methodological approach provided light

into future experimental designs and methodologies to be applied to, and what requirements new experiments pursuing your aim must accomplish.

The last paragraph of the discussion section was edited to highlight the needs in the forthcoming studies: "This study was the first one to address the relationship between monoterpene synthase activities, storage pools and emissions *in situ* in a resin-storing conifer, Scots pine. Our results emphasize the seasonal dynamics and developmental and intraspecific variability in monoterpene biosynthesis and storage, and call for more studies to reveal their connections with emissions. As the emission rates depend on both physiological, structural and environmental factors in a complex manner, future studies should pay attention to the sufficient seasonal coverage of measurements, representation of different aged tissues as well as the number of plant individuals. In particular for improving the models on emission rates, better understanding of these linkages is crucially needed."

Specific comments

P3L14. May be the introduction would gain introducing the role of biotic induced responses as a source of plasticity in the amount and profile of terpenes. Such information would be valuable later in the discussion, as wounds made with the chamber could be a source of emission variability between samplings and plants.

Stress factors are indeed important drivers for monoterpene emissions, and they are briefly discussed in the introductory and discussion sections. However, extending this topic further would lead the manuscript to focus on a different scope than is the main aim. Further, since the chambers were installed with extreme care and used very gently, as discussed in the text, it is not likely that any wounding effects would be seen in our data.

P4L15. You report that you tested the four experimental trees before. So, could you explain what was the reason for sampling in those 4 trees with so different emission spectrum instead to increase your sample size focussing your effort in more similar individuals.

Accessibility was the main reason to select these trees: they grew new to a scaffolding tower and thus their canopies were accessible easily and repeatedly.

P4L25. Please report the mass (mg) of Tenax and Carbopack in the traps for allow experiment repetition.

The amounts of adsorbents (200 and 130 mg, respectively) were added in the text.

P5L5. I would suggest reportting the solvent:sample (d.w.) ratio for a complete description.

The text was edited as follows: "The analysis of monoterpene storage pools followed the methods introduced in Fischbach et al. (2000, 2002) and further developed in Ghirardo et al. (2010). 1 ml of pentane was added as a solvent to 50 mg of frozen needles. The following changes were made to the methods: after pentane extraction --". Thus, we used 18.5 or 24 mg of dried newly formed needles or all other needles, respectively, per 1 ml of solvent.

P5L15. Please, state here the sample size for MTS activity.

The numbers of collected needle samples (= needles for MTS activity and storage pool) are presented in Table 1. Each needle sample consists of several needles as stated in the sampling section.

P6L5. May be I am missing something, but I cannot understand how do you apply this equation because you explain that air entering in the chamber was flowed thru a charcoal trap.

As there is the possibility that charcoal did not remove all the VOCs, also the concentration in the entering air was defined and taken into account in the flux calculation. This is a standard procedure applied in chamber measurements and presented e.g. in Tarvainen et al (2005) and Hakola et al (2006).

P6L10. Please, report what was the range of temperatures for your emission samplings.

The range was added: "The air temperature ranged between –3.5°C (March 2009) and 22.4 (May 2010) during the emission samplings."

P21. (fig 5) In order to be consistent with the codes in other figures and in the text, suggest labelling the panels with the age of the needles instead the actual year. I mean "<>1 yr old" instead "2008 needles".

We agree that consistency is indeed very important for the readers and have tried to improve that throughout the text. Here, we use '2008 and 2009 needles' on purpose. Otherwise there would be only >1-yr-old needles in the B panel of Fig. 5. Thus, no changes in the figure were made.

References (other mentioned references are included in the manuscript reference list)

Kivimäenpää M, Narantsetseg M, Ghimire R, Markkanen J-M, Heijari J, Vuorinen M, Holopainen JK (2012) Influence of tree provenance on biogenic VOC emissions of Scots pine (*Pinus sylvestris*) stumps. Atmospheric Environment 60: 477–485.

[revised manuscript text omitted]

---

## Author Response (AR1)

**Author response**

on the manuscript "Long-term dynamics of monoterpene synthase activities, monoterpene storage pools and emissions in boreal Scots pine" in Biogeosciences Discuss., https://doi.org/10.5194/bg-2018-17-RC1, 2018, by Anni Vanhatalo et al.

Dear Editor.

We have now revised the manuscript according your feedback. We reinforced the discussion part on the potential impacts of substrate availability on monoterpene emission rates and storage pools. The edits are visible in red on pages 12 and 13 of the manuscript. The other edits are responses to the reviewers on the previous review round. We hope the improvements of the manuscript are satisfactory, and the manuscript would now be acceptable for publication in Biogeosciences.

With kind regards on behalf of the co-authors,

Anni Vanhatalo

[revised manuscript text omitted]
, means that the large pool of the substrates, as suggested from other studies (Nogués et al., 2006) is not likely to be the limiting factor of monoterpene emissions. This is valid at least until the photosynthesis is not impaired, as in the four trees of our study. If some of the trees would have been heavily stressed e.g. by herbivory, the low substrate level could have been visible in the MTS activities. However, monoterpene substrate levels were not defined from our samples, so the role of substrate limitation can only be hypothesized.~~

As we aimed at a measurement setup with repeated samplings, the challenge was to maintain the extremely delicate monoterpene storage pools as intact as possible and to avoid any induced emission responses (Ghirardo et al., 2010; Niinemets et al., 2011). However, the monoterpene synthase activities varied between and within trees, and some days with exceptionally high MTS activities and emissions were observed. They may be due to certain short-term processes such as transient responses

to herbivory or mechanical stress. It is known that handling a pine shoot causes increased monoterpene emissions for a few days (Ruuskanen et al., 2005). In addition, the observed anomalous emission blend of tree #3 in September 2009 could originate from a stress reaction, and such unexplained high emission peaks have also been reported earlier in similar measurement setups (e.g. Tarvainen et al., 2005). Nevertheless, no visible damage was observed in the needles prior to sampling, and the twigs were not mechanically injured.

Furthermore, we collected the first set of youngest needles in mid-July, at a time when shoot lignification had not been fully completed and needle elongation was still in progress (ca 85% of final length, see Fig. 2). In practice, sampling short shoots before their lignification is already advanced and causes large wounds and ample resin flow from the wounded twig.

One caveat in the emission measurement method is that the enclosures contained needles from two older year classes, but did not contain the youngest (2009) needles at all. Thus, we cannot separate the effect of needle development and maturation from this data set: almost all emission measurements were performed from already mature needles, and only a fraction of the emissions originated from <1-yr-old needles in the spring and summer of 2009. A part of the emissions may also have originated from non-needle sources, i.e. the woody tissues of twigs.

This study was the first one to address the relationship between monoterpene synthase activities, storage pools and emissions *in situ* in a resin-storing conifer, Scots pine. Our results emphasize the seasonal dynamics and developmental and intraspecific variability in monoterpene biosynthesis and storage, and call for more studies to reveal their connections with emission ratess. As the monoterpene emissions rates depend on both physiologicalphysiological, structural, and and environmental factors andas well as on plant chemotypes in a complex manner, future studies should pay more attention to catch the seasonality as well as the tree-to-tree variation, by using a the sufficient number of measurements and plant individuals. seasonal coverage of measurements, representation of different aged tissues as well as the number of plant individualsAlso, 
[revised manuscript text omitted]